# ABCA1/ABCB1 Ratio Determines Chemo- and Immune-Sensitivity in Human Osteosarcoma

**DOI:** 10.3390/cells9030647

**Published:** 2020-03-06

**Authors:** Dimas Carolina Belisario, Muhlis Akman, Martina Godel, Virginia Campani, Maria Pia Patrizio, Lorena Scotti, Claudia Maria Hattinger, Giuseppe De Rosa, Massimo Donadelli, Massimo Serra, Joanna Kopecka, Chiara Riganti

**Affiliations:** 1Department of Oncology, University of Torino, Via Santena 5/bis, 10126 Torino, Italy; dimascarolina.belisario@unito.it (D.C.B.); muhlis.akman@unito.it (M.A.); martina.godel@edu.unito.it (M.G.); joanna.kopecka@unito.it (J.K.); 2Department of Pharmacy, University of Napoli Federico II, Via D. Montesano 49, 80131 Napoli, Italy; virginia.campani@unina.it (V.C.); lorena.scotti@unina.it (L.S.); gderosa@unina.it (G.D.R.); 3IRCCS Istituto Ortopedico Rizzoli, Laboratory of Experimental Oncology, Pharmacogenomics and Pharmacogenetics Research Unit, Via di Barbiano, 1/10, 40136 Bologna, Italy; mariapia.patrizio@ior.it (M.P.P.); claudia.hattinger@ior.it (C.M.H.); massimo.serra@ior.it (M.S.); 4Department of Neurosciences, Biomedicine and Movement Sciences, Section of Biochemistry, University of Verona, Piazzale L.A. Scuro 10, 37134 Verona, Italy; massimo.donadelli@univr.it

**Keywords:** osteosarcoma, ABCB1, ABCA1, doxorubicin resistance, Vγ9Vδ2 T-cells

## Abstract

The ATP Binding Cassette transporter B1 (ABCB1) induces chemoresistance in osteosarcoma, because it effluxes doxorubicin, reducing the intracellular accumulation, toxicity, and immunogenic cell death induced by the drug. The ATP Binding Cassette transporter A1 (ABCA1) effluxes isopentenyl pyrophosphate (IPP), a strong activator of anti-tumor Vγ9Vδ2 T-cells. Recruiting this population may represent an alternative strategy to rescue doxorubicin efficacy in ABCB1-expressing osteosarcoma. In this work, we analyzed how ABCA1 and ABCB1 are regulated in osteosarcoma, and if increasing the ABCA1-dependent activation of Vγ9Vδ2 T-cells could be an effective strategy against ABCB1-expressing osteosarcoma. We used 2D-cultured doxorubicin-sensitive human U-2OS and Saos-2 cells, their doxorubicin-resistant sublines (U-2OS/DX580 and Saos-2/DX580), and 3D cultures of U-2OS and Saos-2 cells. DX580-sublines and 3D cultures had higher levels of ABCB1 and higher resistance to doxorubicin than parental cells. Surprisingly, they had reduced ABCA1 levels, IPP efflux, and Vγ9Vδ2 T-cell-induced killing. In these chemo-immune-resistant cells, the Ras/Akt/mTOR axis inhibits the ABCA1-transcription induced by Liver X Receptor α (LXRα); Ras/ERK1/2/HIF-1α axis up-regulates ABCB1. Targeting the farnesylation of Ras with self-assembling nanoparticles encapsulating zoledronic acid (NZ) simultaneously inhibited both axes. In humanized mice, NZ reduced the growth of chemo-immune-resistant osteosarcomas, increased intratumor necro-apoptosis, and ABCA1/ABCB1 ratio and Vγ9Vδ2 T-cell infiltration. We suggest that the ABCB1*^high^*ABCA1*^low^* phenotype is indicative of chemo-immune-resistance. We propose aminobisphosphonates as new chemo-immune-sensitizing tools against drug-resistant osteosarcomas.

## 1. Introduction 

The ATP Binding Cassette transporter B1 (ABCB1; P-glycoprotein, Pgp) determines the resistance to a broad spectrum of chemotherapeutic drugs and targeted therapies [1]. In osteosarcoma, the presence of ABCB1 is predictive of poor response to chemotherapy [2,3,4]. Indeed ABCB1 effluxes doxorubicin [1], which, together with cisplatin and methotrexate, is the first-line treatment in patients affected by this tumor [5]. The active efflux of doxorubicin by ABCB1 is the main mechanism of resistance to doxorubicin in osteosarcoma [2,3,4], limiting the drug’s intracellular accumulation and cytotoxicity.

In chemosensitive cells, doxorubicin kills cancer cells via direct mechanisms, i.e., by inhibiting topoisomerase II, increasing reactive oxygen and nitrogen species, perturbing mitochondria functions [6], and indirect mechanisms, i.e., by increasing the ability of the host immune system to eradicate the target cells, producing an immunogenic cell death (ICD) [7]. In particular, doxorubicin elicits the translocation of calreticulin from the endoplasmic reticulum to the plasma membrane. Here, calreticulin triggers the phagocytosis of dying tumor cells by dendritic cells and the subsequent expansion of autologous CD8^+^T-lymphocytes with anti-tumor activity [8]. Both the direct killing effects and the ICD are impaired in ABCB1-expressing cells. First, ABCB1 limits the intracellular concentration of doxorubicin necessary to induce cell death. Second, ABCB1 inhibits the immune-sensitizing function of calreticulin [9,10]. The pharmacological inhibitors of ABCB1 did not reach satisfactory clinical efficacy, because of poor specificity and high toxicity. Targeting signaling pathways up-regulating ABCB1 is an attractive alternative to inhibit this undruggable protein. For instance, lowering cholesterol and isoprenoid-dependent pathways decreases ABCB1 and produces chemo-immune-sensitization in different solid tumors [9,10].

Since doxorubicin-resistant/ABCB1-expressing cells are also immune-resistant, strategies alternative to the canonical ICD are needed to overcome the simultaneous immune-resistance of these cells. With this goal in mind, we focused on the possible exploitation of Vγ9Vδ2 T-cells, a subset that plays a key role in anti-tumor immunity [11]. We would like to investigate if they can represent an endogenous immune-weapon against ABCB1-expressing osteosarcoma cells that escape the classical CD8^+^T-lymphocyte-mediated immune-killing.

γδ T-cells are less than 10% of circulating T-lymphocytes, but they are abundant within the tissues [12]. For this reason, they are physiologically involved in the mucosae defense from microbial pathogens, whereas they induce a strong pro-inflammatory response [13]. On the other hand, a prolonged activation of tissue-infiltrating γδ T-cells favors the break of immune tolerance and the onset of autoimmune diseases [13]. γδ T-lymphocytes peculiarly recognize non-major histocompatible complex (MHC) antigens, structurally characterized by a phosphate moieties (phosphoantigens) [14]. They mount a fast immune response, with intermediate features between innate and adaptive immunity [12,15]. Different subsets of γδ T-cells, producing different cytokines, may have pro-tumor or anti-tumor effects [12,14]. However, it is generally recognized that a high infiltration of Vγ9Vδ2 T-cell subset is a good prognostic factor in solid tumors [16]. When the anti-tumor efficacy of αβ CD8^+^ T-cells is impaired, activated Vγ9Vδ2 T-cells may represent one of the most relevant population in the immune-eradication of cancer cells. Since ABCB1-expressing tumors escape from doxorubicin-induced ICD and CD8^+^T-cell-killing [9,10], we focused our attention on the possibility to rescue the immunogenic death produced by doxorubicin, by increasing the activation of Vγ9Vδ2 T-cells against resistant tumors.

A strong endogenous activator of Vγ9Vδ2 T-cells is isopentenyl pyrophosphate (IPP), an isoprenoid metabolite produced during cholesterol synthesis [17]. We recently identified ATP Binding Cassette transporter A1 (ABCA1), a protein physiologically involved in cholesterol efflux and assembly of nascent HDL [18], as the efflux transporter of IPP in antigen-presenting cells, bone marrow stromal cells, and hematopoietic cells [19]. Moreover, we demonstrated that aminobisphosphonates such as zoledronic acid, a strong inhibitor of farnesyl pyrophosphate synthase (FPPS) [20], increase the intracellular accumulation and extracellular release of IPP in dendritic cells [17,19], activating Vγ9Vδ2 T-lymphocytes with anti-tumor activity.

Until now, sporadic observations correlated ABCA1 to pro-tumor or tumor-suppressive functions in solid cancers [21]. It is not known in solid tumors if cholesterol homeostasis controls ABCA1 expression, if aminobisphosphonates enhance the release of IPP via ABCA1, or if boosting the ABCA1-dependent activation of Vγ9Vδ2 T-cells could be an effective strategy against ABCB1-expressing tumors, resistant to doxorubicin.

To address this question, we examined the expression levels and the molecular circuitries regulating ABCA1 in human osteosarcoma cells with different levels of ABCB1. We found that ABCB1 and ABCA1 are inversely expressed in osteosarcoma cells. Inhibiting the production of FPP, i.e., an intermediate isoprenoid between IPP and cholesterol simultaneously up-regulates ABCA1 and down-regulates ABCB1, acting as a chemo-immune-sensitizer of doxorubicin-resistant cells.

## 2. Materials and Methods

### 2.1. Chemicals

Fetal bovine serum (FBS) and culture medium were purchased from Invitrogen Life Technologies (Carlsbad, CA, USA). Plasticware for cell cultures was purchased from Falcon (Becton Dickinson, Franklin Lakes, NJ, USA). The protein content in cell extracts was assessed with the BCA kit from Sigma-Merck (St. Louis, MO, USA). Electrophoresis reagents were obtained from Bio-Rad Laboratories (Hercules, CA, USA). Doxorubicin and Everolimus (RAD001) were purchased by Sigma-Merck. Self-assembling nanoparticles encapsulating zoledronic acid (NZ) were prepared and characterized as reported in [22,23].

### 2.2. Cells

Human doxorubicin-sensitive osteosarcoma U-2OS and Saos-2 cell lines were purchased from ATCC (Manassas, VA, USA). The corresponding doxorubicin-resistant variants (U-2OS/DX580 and Saos-2/DX580), selected by culturing parental cells in medium containing progressively increasing doxorubicin dosages were generated as reported in [24]. Cells were continuously cultured in the presence of 580 ng/mL doxorubicin. To set 3D cultures of U-2OS and Saos-2 cells, 1 × 10^5^ cells were seeded in 96-well plates coated with Biomimesys™ matrix (Celenys, Rouen, France) [25] and used 7 days after seeding. Cells were monitored by a contrast phase Leica DC100 microscope (Leica Microsystems GmbH, Wetzlar, Germany). All cell lines were authenticated by microsatellite analysis, using the PowerPlex kit (Promega Corporation, Madison, WI, USA; last authentication: June 2019). Cells were maintained in medium supplemented with 10% v/v FBS, 1% v/v penicillin-streptomycin, 1% v/v L-glutamine.

### 2.3. Flow Cytometry

Cells were harvested, washed once in PBS, twice with 10 mM Hepes in Hank’s balanced salt solution, and fixed with 4% v/v paraformaldehyde in PBS for 5 min. After a washing step with Hepes, cells were permeabilized with 0.1% w/v saponin and incubated with anti-ABCB1 (clone MRK16; Kamiya, Tukwila, WA, USA, dilution 1/100) or anti-ABCA1 (HJI, Abcam, Cambridge, UK, dilution 1/100) antibodies. After washing with saponin, cells were incubated with a secondary Alexa488-conjugated antibody (Abcam), washed twice with saponin, and once with Hepes. In the negative control, the primary antibody was replaced by 0.1% w/v saponin. The results were analyzed with a Guava® easyCyte flow cytometer (Millipore, Billerica, MA, USA), equipped with the InCyte software (Millipore).

### 2.4. Cell Viability

Cell viability was measured by the ATPlite Luminescence Assay System (PerkinElmer, Waltham, MA, USA), as per manufacturer’s instructions, using a Synergy HT Multi-Detection Microplate Reader (Bio-Tek Instruments, Winooski, VT, USA) to measure the relative luminescence units (RLU). The RLUs of untreated cells was considered as 100% viability; the results were expressed as a percentage of viable cells versus untreated cells.

### 2.5. Synthesis of Cholesterol, FPP, and IPP

Cells were labeled with 1 µCi of [^3^H]-acetate (3600 mCi/mmol; Amersham International, Piscataway, NJ, USA) for 24 h. The synthesis of radiolabeled cholesterol, FPP [26], and IPP [17] were measured after lipid extraction, separation by thin-layer chromatography (TLC), and liquid scintillation count. Results were expressed as pmoles/mg cell proteins, according to the relative calibration curves.

### 2.6. Release of Cholesterol and IPP

To measure the efflux of cholesterol or IPP, 1 × 10^6^ cells/ml were labeled for 1 h with a pulse of 0.02 mCi of [^14^C]-cholesterol (60 mCi/mmol; Amersham International) [27] or [^14^C]-IPP (50 mCi/mmol; Amersham International) [17], washed five times with PBS, and left for 24 h in fresh medium. After this incubation time, supernatants were collected, lipids were extracted, processed by TLC, and liquid scintillation was used to measure the effluxed cholesterol or IPP. Results were expressed as pmoles/ml, according to the relative calibration curves.

### 2.7. ABCA1 Silencing and Over-Expression

Cells (2 × 10^5^) were transfected with a non-targeting (scrambled) shRNA plasmid (TR30021V) or with a mix of 4 unique shRNAs targeting *ABCA1* (TL315036V), with a non-coding (empty) pCMV6-XL4 vector or with a *ABCA1*-expression vector (SC127939) (all from Origene, Rockville, MD, USA), as per manufacturer’s instructions. The efficacy of silencing or over-expression was verified by immunoblotting as detailed below.

### 2.8. Immunoblotting

Cells were lysed in MLB buffer (125 mM Tris-HCl, 750 mM NaCl, 1% v/v NP40, 10% v/v glycerol, 50 mM MgCl2, 5 mM EDTA, 25 mM NaF, 1 mM NaVO_4_, 10 mg/mL leupeptin, 10 mg/mL pepstatin, 10 mg/mL aprotinin, 1 mM phenylmethylsulphonyl fluoride, pH 7.5), sonicated, and centrifuged at 13,000× *g* for 10 min at 4 °C. Proteins (50 μg) were subjected to immunoblotting and probed with the following antibodies: anti-ABCA1 (HJI, Abcam, dilution 1/500), anti-ABCB1 (C219, Novus Biologicals, Littleton, CO, dilution 1/250), anti-phospho(Ser473)Akt (6F5, Millipore, dilution 1/1000), anti-Akt (SKB1, Millipore, dilution 1/500), anti-phospho(Thr389)-p70 S6K (#9205, Cell Signaling, Technology, Danvers, MA, dilution 1/1000), anti-phospho(Thr421/Ser424)-p70 S6K (#9204, Cell Signaling Technology, dilution 1/1000), anti-p70 S6K (#9202, Cell Signaling Technology, dilution 1/1000), anti-phospho(Thr202/Tyr204) ERK1/2 (#9101, Cell Signaling Technology, dilution 1/1000), anti-ERK1/2 (137F5, Cell Signaling Technology, dilution 1/1000), anti-Hypoxia Inducible Factor-1α (HIF-1α; 54/HIF-1α, BD, dilution 1/500), followed by a peroxidase-conjugated secondary antibody. Anti-β-tubulin antibody (sc-5274, Santa Cruz Biotechnology Inc., Santa Cruz, CA, USA dilution 1/1000) was used as control of equal protein loading. The proteins were detected by enhanced chemiluminescence (Bio-Rad Laboratories). The Ras guanosina trifosfato (GTP)-bound fraction, taken as an index of prenylated and active Ras, was measured using a pull-down assay with the Raf-1-GST fusion protein-agarose beads-conjugates (Millipore). The immunoprecipitated samples were probed with an anti-Ras antibody (Ras10, Millipore, dilution 1/500). To assess HIF-1α phosphorylation, the whole-cell lysate was immunoprecipitated with an anti-HIF-1α (3C144, Santa Cruz Biotechnology Inc., dilution 1/100), then probed with a biotin-conjugated anti-phosphoserine antibody (AB1603, Sigma-Merck, dilution 1/1000), followed by polymeric streptavidin-horseradish peroxidase-conjugates (Sigma-Merck, dilution 1/10000). To evaluate Liver X Receptor α (LXRα) and HIF-1α nuclear translocations, nuclear extracts were prepared using the Nuclear Extract Kit (Active Motif, La Hulpe, Belgium). Nuclear proteins (10 μg) were resolved by SDS-PAGE and probed with anti-LXRα (61175, Active Motif, dilution 1/500) or anti-HIF-1α antibodies. An anti-TFIID/TATA box-binding protein (TBP) antibody (58C9, Santa Cruz Biotechnology Inc., dilution 1/250) was used as control of equal protein loading.

### 2.9. Vγ9δ2 T-Lymphocytes Induced-Cytotoxicity

Peripheral blood samples were obtained from healthy blood donors; the samples were provided by the local Blood Bank (Fondazione Strumia, AOU Città della Salute e della Scienza, Torino). After the isolation on a Ficoll-Hypaque density gradient, peripheral blood mononuclear cells (PBMC) were subjected to immuno-magnetic sorting with the TCRγ/δ^+^T Cell Isolation Kit (Miltenyi Biotec., Bergisch Gladbach, Germany). The phenotypic characterization of Vγ9δ2 T-lymphocytes was confirmed by staining 5 × 10^5^ isolated cells with anti-TCR Vγ9 (clone B6, BD, dilution 1/50) and anti-CD3 (clone BW264/56, Miltenyi Biotec, dilution 1/10) antibodies [17]. Cells were counted with a Guava® easyCyte flow cytometer (Millipore), equipped with the InCyte software (Millipore). Samples with >80% Vγ9^+^/CD3^+^ cells were included and incubated 48 h with 1 μM zoledronic acid and 10 IU/ml IL-2 (Eurocetus, Milan, Italy), to expand Vγ9δ2 T-lymphocytes [17]. Vγ9δ2 T-lymphocyte killing was measured according to [28], with minor modifications. Vγ9δ2 T-lymphocytes (5 × 10^5^) were cultured overnight with target cells at a 1:1 ratio. After this co-incubation, the supernatant containing Vγ9δ2 T-lymphocytes was removed, while adherent (i.e., osteosarcoma) cells were washed twice with PBS, detached with gentle scraping, and stained with the Annexin V/Propidium Iodide kit (APOAF, Sigma-Merck), as per manufacturer’s instruction. The fluorescence was acquired using a Guava® easyCyte flow cytometer and InCyte software. The percentage of Annexin V^+^/Propidium Iodide^+^ osteosarcoma cells was considered an index of Vγ9δ2 T-lymphocyte killing. The results were expressed as a killing fold change, i.e., percentage of Annexin V^+^/Propidium Iodide^+^ cells in each experimental conditions/percentage of Annexin V^+^/Propidium Iodide^+^ U-2OS or Saos-2 untreated cells.

### 2.10. Chromatin Immunoprecipitation (ChIP)

ChIP samples were prepared as described [27], using ChIP-tested anti-LXRα (61175, Active Motif, dilution 1/50) or anti-HIF-1α (ab2185, Abcam, dilution 1/50) antibodies. The putative Liver X Receptor Response Element (LRE) site on *ABCA1* promoter and Hypoxia Response Element (HRE) on *ABCB1* promoter were validated with the Matinspector Software (https://www.genomatix.de/matinspector.html). Primer sequences were: for *ABCA1* promoter (LRE): 5’-GGAGAGCACAGGCTTTGACC-3’; 5’-CTCTCGCGCAATTACGGG-3’; for *ABCB1* promoter (HRE): 5’-GGAGCAGTCATCTGTGGTGA-3’; 5’-CTCGAATGAGCTCAGGCTTC-3’. Primers used as negative internal controls for a nonspecific 10,000-bp upstream sequence were: 5’-GTGGTGCCTGAGGAAGAGAG-3’; 5’-GCAACAAGTAGGCACAAGCA-3’. The immunoprecipitated products were amplified by qRT-PCR.

### 2.11. qRT-PCR

Total RNA was extracted and reverse-transcribed using an iScript^TM^ cDNA Synthesis Kit (Bio-Rad Laboratories). The qRT-PCR was performed with the IQ SYBR Green Supermix (Bio-Rad Laboratories). The primer sequences, which were designed using the qPrimerDepot database (http://primerdepot.nci.nih.gov/), were: *ABCA1*: 5’-CAGAGCTCACAGCAGGGAC-3’; 5’-CTTCTCCGGAAGGCTTGTC-3’; *ABCB1*: 5’- TGCTGGAGCGGTTCTACG-3’; 5’-ATAGGCAATGTTCTCAGCAATG-3’; *S14*: 5’-CGAGGCTGATGACCTGTTCT-3’; 5’-GCCCTCTCCCACTCTCTCTT-3’. The relative quantification was performed by comparing each PCR product with the housekeeping PCR product *S14*, using the Bio-Rad Software Gene Expression Quantitation (Bio-Rad Laboratories).

### 2.12. ABCA1 and ABCB1 mRNA Expression on Clinical Samples

Total RNA from 21 high-grade osteosarcoma snap-frozen specimens collected at diagnosis was extracted with TRIzol (Invitrogen). Before extraction, all samples were histologically examined for tissue quality, in order to isolate RNA only from representative specimens. Fragmentation of cRNA, hybridization to the Affymetrix hg-u133, plus 2.0 microarrays and scanning were performed as described by [29]. After normalization of expression data with the MAS5.0 algorithm, gene expression profiles were analyzed by using the freely available R2 web application and visualization platform (http://r2.amc.nl). Correlation of mRNA gene expression with clinical outcome was assessed by evaluating the relapse-free survival curves. Overall survival was not considered because of the heterogeneity of post-relapse treatments. For survival analyses, patients were stratified by using, as a cut-off, the median expression level of each transporter to distinguish low-expressing (expression level lower than the median value) from high-expressing patients (expression level equal or higher than the median value). Kaplan–Meier and log-rank methods were used to draw and evaluate the significance of survival curves.

### 2.13. In Vivo Tumor Growth

U-2OS 3D cells (1 × 10^6^), mixed with 100 μl Matrigel, were injected subcutaneously (s.c.) in female non-obese diabetic (NOD) severe combined immunodeficiency (SCID) gamma mice engrafted with human hematopoietic CD34^+^ cells (Hu-CD34^+^; The Jackson Laboratories, Bar Harbor, MA), a model previously reported to have functional reconstituted VγVδ T-cells [30]. Mice were housed (5 per cage) under 12 h light/dark cycle, with food and drinking provided ad libitum. Tumor growth was measured daily by caliper, according to the equation (LxW^2^)/2, where L = tumor length and W = tumor width. When the tumor reached the volume of 50 mm^3^, animals were randomized and treated (on day 3, 9, and 15 after randomization) as follows: 1) vehicle group, treated with 0.1 ml saline solution intravenously (i.v.); 2) doxorubicin group, treated with 5 mg/kg doxorubicin i.v.; 3) NZ group, treated with 20 μg/mice zoledronic acid as encapsulating nanoparticles i.v.; 4) NZ + doxorubicin group, treated with 20 μg/mice zoledronic acid as NZ and 5 mg/kg doxorubicin i.v. Tumor volumes were monitored by caliper and animals were euthanized at day 21 after randomization with zolazepam (0.2 ml/kg) and xylazine (16 mg/kg). Tumor volume and animal weight were monitored throughout the study. The tumors were then excised and photographed. Animal care and experimental procedures were approved by the Bio-Ethical Committee of the Italian Ministry of Health (#627/2018-PR, 10/08/2018).

### 2.14. Immuno-Histochemical and Lymphocytic Infiltrate Analysis

Tumors were resected and fixed in 4% v/v paraformaldehyde, stained for hematoxylin-eosin, or immunostained for Ki67 (AB9260, Millipore, dilution 1/50), cleaved (Asp175)caspase 3 (#9661, Cell Signaling, Technology, dilution 1/50 ), ABCB1 (Novus Biologicals, dilution 1/50), or ABCA1 (Abcam, dilution 1/50) followed by a peroxidase-conjugated secondary antibody. Nuclei were counter-stained with hematoxylin. Sections were examined with a Leica DC100 microscope. Excised tumors were digested with 1 mg/mL collagenase and 0.2 mg/mL hyaluronidase (1 h at 37 °C) and filtered using a 70 μm cell strainer to obtain a single-cell suspension. Infiltrating immune cells were collected by centrifugation on Ficoll-Hypaque density gradient and subjected to immune phenotyping by flow cytometry, using antibodies against anti-TCR Vγ9 (BD, dilution 1/50) and anti-CD3 (Miltenyi Biotec, dilution 1/10), to detect Vγ9δ2 T-lymphocytes, anti-CD3 and anti-CD4 (M-T466, Miltenyi Biotec, dilution 1/10) for CD4^+^ T-lymphocytes, anti-CD3 and anti-CD8 (BW135/80, Miltenyi Biotec, dilution 1/10) for CD8^+^ T-lymphocytes, and anti-CD19 (REA675 Miltenyi Biotec, dilution 1/10) for B lymphocytes. Positive cells were quantified using a Guava® easyCyte flow cytometer and InCyte software. Results were expressed as a percentage of Vγ9^+^CD3^+^ cells, CD3^+^CD4^+^ cells, CD3^+^CD8^+^ cells, and CD19^+^ cells, over CD3^+^ cells.

### 2.15. Statistical Analysis

All data in the text and figures are provided as means ± SD. The results were analyzed by a one-way analysis of variance (ANOVA), using the Statistical Package for Social Science (SPSS) software (IBM SPSS Statistics v.19). *p* < 0.05 was considered significant.

## 3. Results

### 3.1. The Ratio ABCB1/ABCA1 Increases in Doxorubicin-Resistant Osteosarcoma Cells

We first compared the expression of ABCB1 and ABCA1 in doxorubicin-sensitive U-2OS cells, in the resistant counterpart U-2OS/DX580 (both grown as standard 2D cultures), and in 3D cultures of U-2OS cells (Figure 1A). Compared to 2D U-2OS cells, both U-2OS/DX580 and 3D U-2OS cells had increased levels of ABCB1 on their surface. Surprisingly, the amount of surface ABCA1 was higher in 2D U-2OS cells and lower in U-2OS/DX580 and 3D U-2OS cells (Figure 1B). Consistently, the poorly ABCB1-expressing 2D U-2OS cells had a reduced viability when treated with doxorubicin, while the highly ABCB1-expressing U-2OS/DX580 and 3D U-2OS cells were doxorubicin-resistant (Figure 1C). At the same time, the efflux of cholesterol, the physiological substrate of ABCA1, was higher in 2D U-2OS cells, rich in ABCA1, and lower in U-2OS/DX580 and 3D U-2OS cells (Figure 1D), characterized by low levels of ABCA1.

This phenotype was not cell-line-specific: indeed, doxorubicin-sensitive Saos-2 cells, grown as 2D cultures, showed lower ABCB1, higher ABCA1 (Appendix A), reduced viability in response to doxorubicin (Appendix A), and higher efflux of cholesterol (Appendix A) compared to 2D Saos-2/DX580 and 3D Saos-2 cells.

### 3.2. ABCB1^high^ABCBA1^low^ Phenotype is Suggestive of a Worse Clinical Outcome

Comparison of mRNA gene expression level in osteosarcoma and human normal muscle tissue samples (Appendix A) showed that expression of ABCA1 was significantly higher in osteosarcoma compared either to normal muscle (Asmann series; *p* < 0.001) or to normal skeletal muscle (Gordon series; *p* < 0.05). No significant differences were found concerning the expression level of ABCB1 mRNA in OS versus normal muscle tissues.

Correlation between mRNA gene expression level at diagnosis and outcome was evaluated by considering relapse-free survival (Appendix A). By stratifying patients in high and low-expressors on the basis of the median gene expression level of the whole 21 high-grade osteosarcoma samples series, no statistically significant correlations were found. However, survival curves suggested a trend toward a worse clinical outcome (higher probability to relapse) for patients with lower ABCA1 expression (P = 0.624) or with higher ABCB1 expression levels (P = 0.675).

### 3.3. ABCA1 is a Determinant of Immune-Sensitivity in Osteosarcoma Cells

We next investigated if ABCA1 mediates the efflux of IPP and the anti-tumor activity of Vγ9δ2 T-lymphocytes in osteosarcoma, as it does in dendritic cells [19]. 2D U-2OS cells knocked-down for ABCA1 (Figure 2A) had a reduced efflux of IPP to the same level of 3D U-2OS cells (Figure 2B). Contrarily, 3D U-2OS cells overexpressing exogenous ABCA1 as 2D U-2OS cells (Figure 2A), also increased IPP efflux to the same level of the latter (Figure 2B). This experimental set demonstrates that ABCA1 is directly involved in IPP transport in osteosarcoma cells.

Both 2D U-2OS and 2D Saos-2 cells had higher efflux of IPP than the 2D DX580 sublines and the 3D cultures (Figure 2C; Appendix A). Consistently, 2D U-2OS and 2D Saos-2 cells were significantly more killed by Vγ9δ2 T-lymphocytes than 2D U-2OS/DX580, 3D U-2OS (Figure 2D), 2D Saos-2/DX580, and 3D Saos-2 cells (Appendix A).

The higher efflux of cholesterol and IPP in doxorubicin-sensitive cells was not due to the increased synthesis of these lipids: indeed, 2D U-2OS and 2D Saos-2 cells had lower synthesis of endogenous cholesterol (Appendix A) and IPP (Appendix A) than the DX580 sublines or 3D cultures. This finding suggests that the differential efflux in cholesterol and IPP depends on the different levels of ABCA1 present in each subline.

Overall, these data demonstrate that ABCB1 is high and ABCA1 is low in doxorubicin-resistant osteosarcoma cells. This feature makes drug-resistant cells also immune-resistant, since they are less killed by Vγ9δ2 T-lymphocytes. Second, our data suggest that 3D cultures of U-2OS and Saos-2 cells represent a reliable tool to study the molecular bases of this chemo-immune-resistant phenotype and to test chemo-immune-sensitizing strategies. We thus focused on 3D U-2OS cells in the following experiments.

### 3.4. Ras/Akt/mTOR Axis Negatively Regulates the Transcription of ABCA1 in Resistant Osteosarcoma Cells

FPP is an isoprenoid metabolite downstream IPP, necessary for the isoprenylation and activation of monomeric G-proteins like Ras [31]. PI3K/Akt/mTOR axis is among the Ras-controlled pathways [32]. It down-regulates cholesterol efflux, by reducing the LXRα-driven transcription of *ABCA1* [33]. To manipulate this cascade, we targeted the synthesis of FPP using the aminobisphosphonate zoledronic acid. To this aim, we used NZ, a self-assembling nanoparticle encapsulating zoledronic acid, characterized by a higher uptake in tumor cells and the most favorable pharmacokinetic profile compared to free zoledronic acid [22].

NZ significantly reduced FPP synthesis in chemo-immune-resistant 3D U-2OS cells (Figure 3A), which had a constitutively activated Ras/Akt/mTOR pathway, as suggested by the basally high levels of GTP-bound Ras, phospho(Ser473)Akt, and phospho(Thr389/Thr421/Ser424)S6K (Figure 3B). Consistently, the amount of LXRα translocated into the nucleus, corresponding to the active form, was low in 3D U-2OS cells (Figure 3C). The mTOR inhibitor Everolimus (RAD001) increased the nuclear levels of LXRα, as the specific LXRα activator TO901317 did (Figure 3C), indicating that LXRα activity was negatively regulated by Ras/Akt/mTOR axis in our model. Also, NZ reduced the activity of Ras/Akt/mTOR axis (Figure 3B) and increased the nuclear translocation of LXRα (Figure 3C) in 3D U-2OS cells. Similarly to Everolimus and TO901317, NZ increased the binding of LXRα to the *ABCA1* promoter (Figure 3D) and the levels of ABCA1 mRNA (Figure 3E). Moreover, NZ-, Everolimus-, and TO901317-treated cells showed higher efflux of IPP (Figure 3F) and Vγ9δ2 T-lymphocyte killing (Figure 3G), suggesting that targeting the Ras/Akt/mTOR axis and increasing ABCA1 restore the immune-sensitivity in refractory 3D U-2OS cells.

### 3.5. ERK1/2/HIF-1α Axis up-Regulates ABCB1 in Resistant Osteosarcoma Cells

ERK1/2 is another downstream effector of Ras [32]. In solid and hematologic tumors ERK1/2 phosphorylates HIF-1α on serine, stabilizing it and increasing the HIF-1α-driven transcription of *ABCB1* [23,26,34]. In line with the constitutively active Ras (Figure 3B), 3D U-2OS cells had a basal activation of ERK1/2, indicated by detectable levels of phospho(Thr202/Tyr204)ERK1/2 and phospho(Ser)HIF-1α (Figure 4A). HIF-1α was stably translocated in the nucleus (Figure 4B) and bound to the *ABCB1* promoter (Figure 4C). NZ reversed all these events (Figure 4A–C) and reduced the levels of ABCB1 mRNA (Figure 4D). Thanks to the inhibition of Ras/Akt/mTOR (Figure 3) and Ras/ ERK1/2/ HIF-1α (Figure 4) axes, NZ shifted the expression pattern of ABCA1 and ABCB1 in 3D U-2OS cells, turning them from ABCA1*^low^*ABCB1*^high^* cells into ABCA1*^high^*ABCB1*^low^* cells (Figure 4E). This effect is suggestive of the transition from a chemo-immune-resistant phenotype to a chemo-immune-sensitive one.

### 3.6. Increasing the ABCA1/ABCB1 Ratio Reduces the Growth of Osteosarcoma in Preclinical Models

Since mice have different VγVδ T-cell sets than humans [11], to verify whether NZ could reverse chemo-immune-resistance in preclinical models of osteosarcoma, we used Hu-CD34^+^ mice, characterized by functional Vγ9δ2 T-lymphocytes [30]. Mice were implanted with cells derived by chemo-immune-resistant 3D U-2OS cultures, treated with doxorubicin at the maximally tolerated dose [35], with or without NZ, at a dosage corresponding to that administered to patients treated with zoledronic acid. Doxorubicin was completely ineffective in reducing tumor growth, while NZ slowed it down. The maximal reduction in tumor volume was obtained by the combination of NZ and doxorubicin (Figure 5A,B). None of the group treatments showed significant changes in animal weight (Appendix A). The effect of NZ was pleiotropic: particularly in association with doxorubicin, it increased the amount of intratumor necrotic areas, as observed in hematoxylin-eosin staining, reduced the tumor cell proliferation, indicated by lower positivity for Ki67, and increased the cleaved caspase 3, i.e., the caspase-activated during apoptosis (Figure 5C,D). Moreover, NZ, alone and in association with doxorubicin, reduced the intratumor levels of ABCB1 (Figure 5C,D), potentially increasing the sensitivity to doxorubicin, increased the intratumor expression of ABCA1 (Figure 3C,D) and infiltration of Vγ9δ2 T-lymphocytes (Figure 3E), suggesting an enhanced recruitment of this set of immune cells within the tumor mass. In contrast, CD3^+^CD4^+^ T-lymphocytes, the most abundant lymphocytic infiltrating population, and CD3^+^CD8^+^ T-lymphocytes did not change between the different group of treatments (Appendix A), while B-lymphocytes were very low (always < 1.5% compared to total CD3^+^ T-lymphocytes) within tumor mass (data not shown).

## 4. Discussion

Despite its well-known role as a prognostic factor in different solid tumors [21], no reports documented a clinical role for ABCA1 in osteosarcoma. A very recent analysis of the downstream signaling of parathyroid hormone receptor 1 identified *ABCA1* as a putative gene important in osteosarcoma progression [36]. The role of ABCA1 was attributed to its control of cholesterol homeostasis and efflux, a function that was maintained in the osteosarcoma cell lines examined in the present work.

We found ABCA1 expressed in doxorubicin-sensitive osteosarcoma cells, in line with previous findings showing a constitutive expression of ABCA1 in doxorubicin-sensitive triple-negative breast cancer cells [37]. Unexpectedly, in osteosarcoma cells with acquired resistance to doxorubicin and high levels of ABCB1, the amount of ABCA1 was reduced. This phenotype was observed also in 3D cultures of osteosarcoma derived from doxorubicin-sensitive cells, which up-regulated ABCB1 and displayed resistance to doxorubicin, as it occurs in colon [25] and breast [38] cancer cells. We cannot exclude that the reduction of ABCA1 plays a role in chemoresistance, together with the increased level of ABCB1. Indeed, a reduced efflux of cholesterol in osteosarcoma doxorubicin-resistant cells, that are characterized by low levels of ABCA1 and high synthesis of cholesterol, may determine an increased accumulation of cholesterol in the plasma membrane. This condition favors the activity of ABCB1 [39,40].

In a limited series of high-grade osteosarcomas, a non-significant trend of the high ABCB1 mRNA levels or the low ABCA1 mRNA levels at diagnosis toward a worse relapse-free survival. This observation is in agreement with our previously reported body of evidence, which demonstrated that in high-grade osteosarcoma patients only ABCB1 protein expression level at clinical onset was related to an unfavorable prognosis [2,3,41,42] and reference therein. The prognostic meaning of ABCA1 and its physiological or pathological function in osteosarcoma has never been studied.

In the present work, we focused on a recently-discovered function of ABCA1, i.e., the efflux of the isoprenoid metabolite IPP that activates Vγ9Vδ2 T-cells [19], a subset of T-lymphocytes with potential killing activity against transformed cells that act also when αβ T-cell-mediated immunity is impaired [11,12,14]. Our gain- and loss-of-function experiments provided the demonstration that ABCA1 transports IPP and modulates the activity of Vγ9Vδ2 T-lymphocytes in osteosarcoma, as it does in antigen-presenting cells [19]. Moreover, we report for the first time a new chemo-immune-resistant phenotype, based on the simultaneous up-regulation of ABCB1 and down-regulation of ABCA1. 

Since a high production of cholesterol and upstream metabolites, like FPP, favors both the increase of ABCB1 activity [40] and the down-regulation of ABCA1 [19], and as such, we focused on cholesterol-related pathways to decipher the mechanisms underlying this ABCB1*^high^*ABCA1*^low^* phenotype.

Using the 3D cultures of U-2OS cells as a prototypical example of ABCB1*^high^*ABCA1*^low^* cells, characterized by strong resistance to doxorubicin and to Vγ9Vδ2 T-cell killing, we identified the Ras/Akt/mTOR axis as a down-regulator of ABCA1, through its inhibition on LXRα, and the Ras/ERK1/2/HIF-1α axis as an up-regulator of ABCB1. Both Akt/mTOR and ERK1/2 axes mediate cell proliferation, migration, and inhibition of apoptosis in osteosarcoma [43,44,45]. As far as drug resistance is concerned, Akt/mTOR induces resistance to doxorubicin, cisplatin, and methotrexate by promoting pro-survival autophagy [46]. Consistently, Everolimus has been proposed as a second-line treatment in osteosarcoma: unluckily, it did not achieve a complete response in patient-derived doxorubicin-resistant osteosarcoma xenografts [47] or in patients with high-grade relapsed osteosarcoma [48] when used alone. The reduction of tumor growth in xenografts and the increase in the progression-free survival of patients was achieved only when Everolimus was combined with Sorafenib [47,48]. These results suggest that targeting only the Akt/mTOR axis is not sufficient to induce a robust anti-tumor effect and/or to chemosensitize refractory osteosarcoma. ERK1/2 also promotes drug resistance by favoring osteosarcoma cell proliferation and preventing the apoptosis induced by DNA-damaging agents such as cisplatin [49]. HIF-1α has been reported to have a direct linkage with the ABCB1-mediated chemoresistance in osteosarcoma: indeed, HIF-1α is induced during the acquisition of doxorubicin resistance by the continuous exposure to the drug and in turn, it mediates the resistance to doxorubicin by up-regulating ABCB1 [50].

Notably, both Akt/mTOR and ERK1/2 are controlled by the activity of Ras [32] that depends on the availability of FPP for its activity. In order to decrease at the same time Akt/mTOR and ERK1/2 signaling, we decided to target the synthesis of FPP to turn off Ras activity. In previous work, we demonstrated that zoledronic acid was effective in reducing Ras farnesylation and Akt/mTOR activation in human and murine osteosarcoma cells [51]. Here, we refined the zoledronic administration using self-assembled nanoparticles (NZ) containing the drug, characterized by a higher half-life in blood and higher delivery into the tumor compared to non-transformed bone [22].

On the one hand, NZ down-regulated Ras/Akt/mTOR axis, as Everolimus did, removed the inhibition on LXRα-transcriptional activity, restored ABCA1 expression, and Vγ9Vδ2 T-cell killing. On the other hand, NZ decreased the activation of ERK1/2/HIF-1α and the expression of ABCB1. Most importantly, it reduced the growth of osteosarcoma tumors derived from chemo-immune-resistant cultures that were completely refractory to doxorubicin. The effect of NZ alone could be attributed to the reduction of Ras/Akt/mTOR and Ras/ERK1/2 pro-survival pathways, as well as to the restored activity of Vγ9Vδ2 T-cells. The higher anti-tumor activity of the combination of NZ and doxorubicin can be due to the additional down-regulation of ABCB1 that allowed an increased intratumor retention of doxorubicin and its cytotoxicity.

Given the tropism for bone, aminobisphosphonates, as free drugs or polymeric nanoparticles, have been proposed as therapeutic options for osteosarcoma. For instance, in preclinical models alendronic acid exerted anti-tumor, anti-angiogenic, and anti-metastatic effects [52,53]. Interestingly, zoledronic acid is known to raise the cytotoxic activity of Vγ9Vδ2 T-cells against osteosarcoma cells, in particular against cells pretreated with the anti-Her2 Trastuzumab antibody [54]. Indeed, the immune system is more active against damaged tumor cells [8]. The combined treatment of osteosarcoma with zoledronic acid and chemotherapy like doxorubicin or targeted-therapy like Trastuzumab [54], that damage tumor cells, may explain the higher anti-tumor activity of Vγ9Vδ2 T-cells.

Although the role of γδ T-cells as anti-tumor or pro-tumor population is still controversial, depending on tumor type, γδ T-cells subset, and amount [16,55], Vγ9Vδ2 T-cells are considered the most favorable prognostic factor between the tumor-infiltrating immune populations in 25 different types of solid cancers including osteosarcoma [16]. In our preclinical models, NZ recapitulates what we observed in ex vivo co-culture models: it increased the percentage of necro-apoptotic cells, the ABCA1/ABCB1 ratio and the infiltration of Vγ9Vδ2 T-cells. This feature is indicative of an effective rescue of doxorubicin efficacy, consequent to the reduction of ABCB1, and of an increased Vγ9Vδ2 T-cells killing activity, consequent to an increase in ABCA1. Together with a decreased proliferation, this phenotype may explain the marked reduction in tumor growth in animals treated with NZ and doxorubicin. The increased tumor-infiltration of Vγ9Vδ2 T-cells achieved by NZ should be considered as the starting point for designing new strategies of adoptive immune-therapy for osteosarcoma, as already experimented in clinical trials for other tumors [56], in particular, if refractory to the anti-tumor immune activity of CD8^+^T-lymphocytes. We did not detect any changes in CD4^+^ and CD8^+^T-lymphocytes infiltrating the tumors upon the treatment with NZ, suggesting a peculiarity of the drug for γδ T-cells. This result is in line with previous findings showing that Vγ9Vδ2 T-cells infiltrating the tumors may vary independently from other T-lymphocyte subsets [57]. Indeed Vγ9Vδ2 T-cells recognize different antigens and mount a different immune-response from CD4^+^ and CD8^+^T-lymphocytes [12,14,15].

Moreover, our findings may indicate new treatment modalities based on a different use of zoledronic acid, which was already included as an adjuvant to chemotherapy to treat osteosarcoma patients in the OS2006 phase III study (NCT00470223), but did not provide improvement of treatment efficacy and prognosis [58].

## 5. Conclusions

In conclusion, we demonstrated that ABCB1 and ABCA1 are inversely expressed in doxorubicin-sensitive and doxorubicin-resistant osteosarcoma cells. The pattern of differential expression makes chemosensitive cells also immune-sensitive, and chemoresistant cells also immune-resistant. Until now, chemo-immune-sensitizing treatments have been poorly investigated in osteosarcoma. By identifying druggable pathways underlying the ABCB1*^high^*ABCA1*^low^* phenotype, we propose a new treatment option for drug-resistant osteosarcoma cells. Enhancing the anti-tumor activity of endogenous Vγ9Vδ2 T-cells against ABCB1-positive osteosarcoma, where doxorubicin fails, can be a step forward in the immune-therapy-based treatments for aggressive osteosarcomas.

## Figures and Tables

**Figure 1 cells-09-00647-f001:**
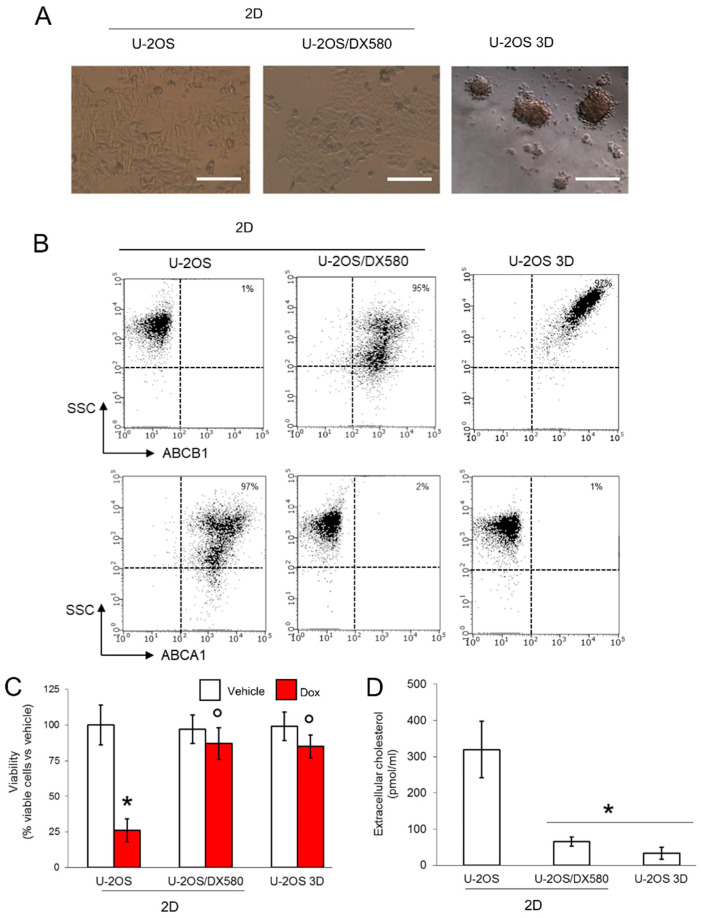
3D cultures of U-2OS cells display an ABCB1*^high^*ABCA1*^low^* phenotype. (**A**) Representative micro-photographs of U-2OS cells, U-2OS/DX580 cells, both grown as 2D cultures, and U-2OS cells grown in 3D culture (10× ocular lens, 4× objective). Bar: 100 μM. (**B**) Dot plots of ABCB1 and ABCA1 proteins on the cell surface, measured by flow cytometry in duplicates. The figure is representative of one out of three experiments. SSC: side scattering. Percentage of ABCB1- and ABCA1-positive cells, calculated as cells with a fluorescence >10 ^2^ using the Incyte software. (**C**) Cells were grown for 72 h in medium containing 5 μM DMSO (vehicle) or doxorubicin (Dox). Percentage of viable cells, measured by a chemiluminescence-based assay in quadruplicates. Data are means ± SD (n = 4 independent experiments). * *p* < 0.001 for doxorubicin-treated cells vs. untreated cells; ° *p* < 0.001 for doxorubicin-treated 2D U-2OS/DX580 and 3D U-2OS cells vs. doxorubicin-treated 2D U-2OS cells. (**D**) Cells were labeled 1 h with [^14^C]-cholesterol and extensively washed. After 24 h, the [^14^C]-cholesterol collected in the supernatant, considered an index of cholesterol efflux, was measured by liquid scintillation in duplicates. Data are means ± SD (n = 3 independent experiments). * *p* < 0.001 for 2D U-2OS/DX580 and 3D U-2OS cells vs. 2D U-2OS cells. ABCB1: ATP Binding Cassette transporter B1; ACBA1: ATP Binding Cassette transporter A1.

**Figure 2 cells-09-00647-f002:**
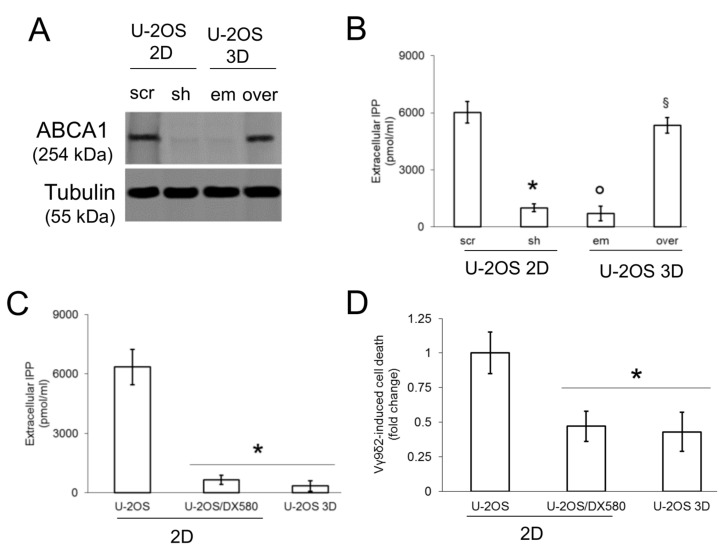
ABCA1 effluxes isopentenyl pyrophosphate (IPP) and mediates the activation of Vγ9δ2 T-lymphocytes against osteosarcoma cells. (**A**) 2D U-2OS cells were transfected with a non-targeting (scrambled, scr) shRNA plasmid or with shRNAs targeting *ABCA1* (sh). 3D U-2OS cells were transfected with an empty (em) vector or with an *ABCA1*-expression vector (over). Forty-eight hours after the transfection, the amount of ABCA1 was verified by immunoblotting. The β-tubulin expression was used as control of equal protein loading. The figure is representative of one out of three experiments. (**B**) Cells were treated as in (**A**). Forty-eight hours after the transfection, cells were labeled 1 h with [^14^C]-IPP and extensively washed. After 24 h, the [^14^C]-IPP collected in the supernatant, considered an index of IPP efflux, was measured by liquid scintillation. Data are means ± SD (n = 3 independent experiments). * *p* < 0.001 for sh 2D U-2OS vs. scr 2D U-2OS cells; ° *p* < 0.001 for em 3D U-2OS vs. scr 2D U-2OS cells; ^§^p<0.001 for over 3D U-2OS vs. em 3D U-2OS cells. (**C**) The IPP efflux was measured in U-2OS cells, U-2OS/DX580 cells, both grown as 2D cultures, and U-2OS cells grown in 3D culture, as detailed at point (**B**), in duplicates. Data are means ± SD (n = 3 independent experiments). * *p* < 0.001 for 2D U-2OS/DX580 and 3D U-2OS vs. 2D U-2OS cells. (**D**) Vγ9δ2 T-lymphocytes were cultured overnight with 2D U-2OS, 2D U-2OS/DX580 and 3D U-2OS cells. The percentage of Annexin V/Propidium Iodide-positive cells was measured by flow cytometry, in duplicates. Data are means ± SD (n=3 independent experiments). * *p* < 0.001 for 2D U-2OS/DX580 and 3D U-2OS vs. 2D U-2OS cells.

**Figure 3 cells-09-00647-f003:**
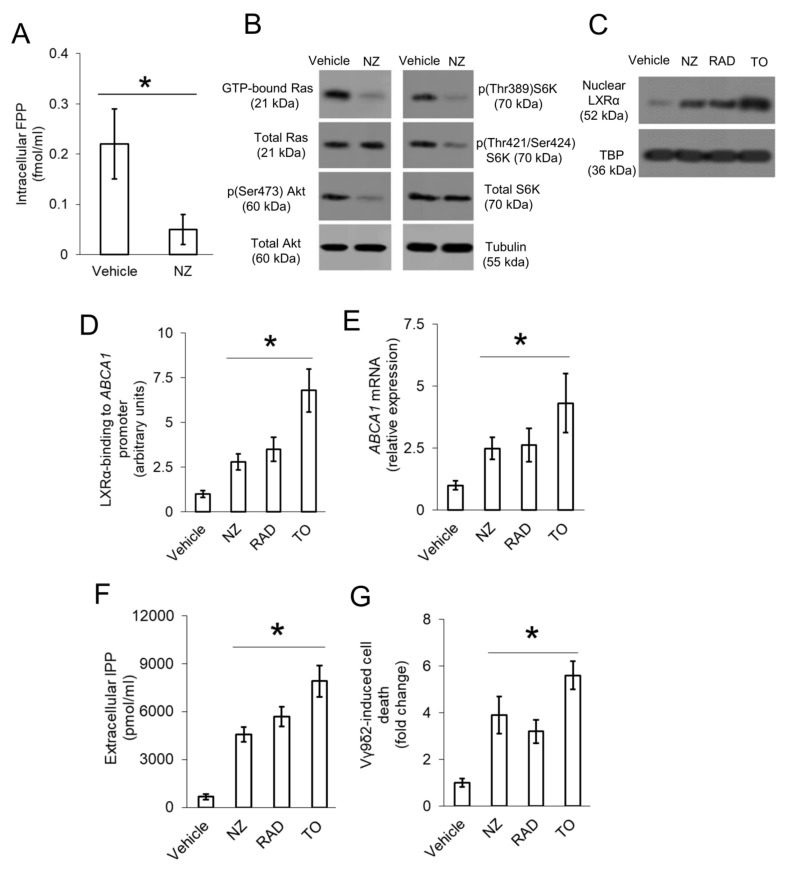
ABCA1 is down-regulated by Ras/Akt/mTOR and restored by Liver X Receptor α (LXRα) activation in resistant osteosarcoma cells. 3D U-2OS cells were treated with sterile physiological solution (vehicle) or self-assembled nanoparticles encapsulating zoledronic acid (NZ), containing 1 μM zoledronic acid for 24 h. When indicated, the mTOR inhibitor Everolimus (RAD001) (RAD; 10 nM) or the LXRα activator TO901317 (TO; 100 nM) were added for 24 h. (**A**) Cells were radiolabeled 24 h with [^3^H]-acetate. The activity of farnesyl pyrophosphate synthase (FPPS), taken as an index of de novo synthesis of [^3^H]-FPP, was measured by thin-layer chromatography (TLC) separation and liquid scintillation, in duplicates. Data are means ± SD (n = 3 independent experiments). * *p* < 0.001 for NZ-treated vs. vehicle-treated cells. (**B**) Pull-down assay of GTP bound-Ras and immunoblotting of the indicated proteins in whole-cell extracts. The β-tubulin expression was used as control of equal protein loading. The figure is representative of one out of three experiments. (**C**) Immunoblotting of LXRα in nuclear extracts. The TBP expression was used as control of equal protein loading. The figure is representative of one out of three experiments. (**D**) LXRα binding to *ABCA1* promoter, measured by chromatin immunoprecipitation (ChIP) in triplicates. Data are means ± SD (n = 3 independent experiments). **p* < 0.001 for all treatments vs. vehicle-treated cells. (**E**) ABCA1 mRNA levels, measured by qRT-PCR in triplicates. Data are means ± SD (n = 3 independent experiments). * *p* < 0.001 for all treatments vs. vehicle-treated cells. (**F**) IPP efflux, measured after metabolic radiolabeling, in duplicates. Data are means ± SD (n = 3 independent experiments). * *p* < 0.001 for all treatments vs. vehicle-treated cells. **G.** Vγ9δ2 T-lymphocyte killing activity, measured as a percentage of Annexin V/Propidium Iodide-positive target cells, detected by flow cytometry, in duplicates. Data are means ± SD (n = 3 independent experiments). * *p* < 0.001 for all treatments vs. vehicle-treated cells.

**Figure 4 cells-09-00647-f004:**
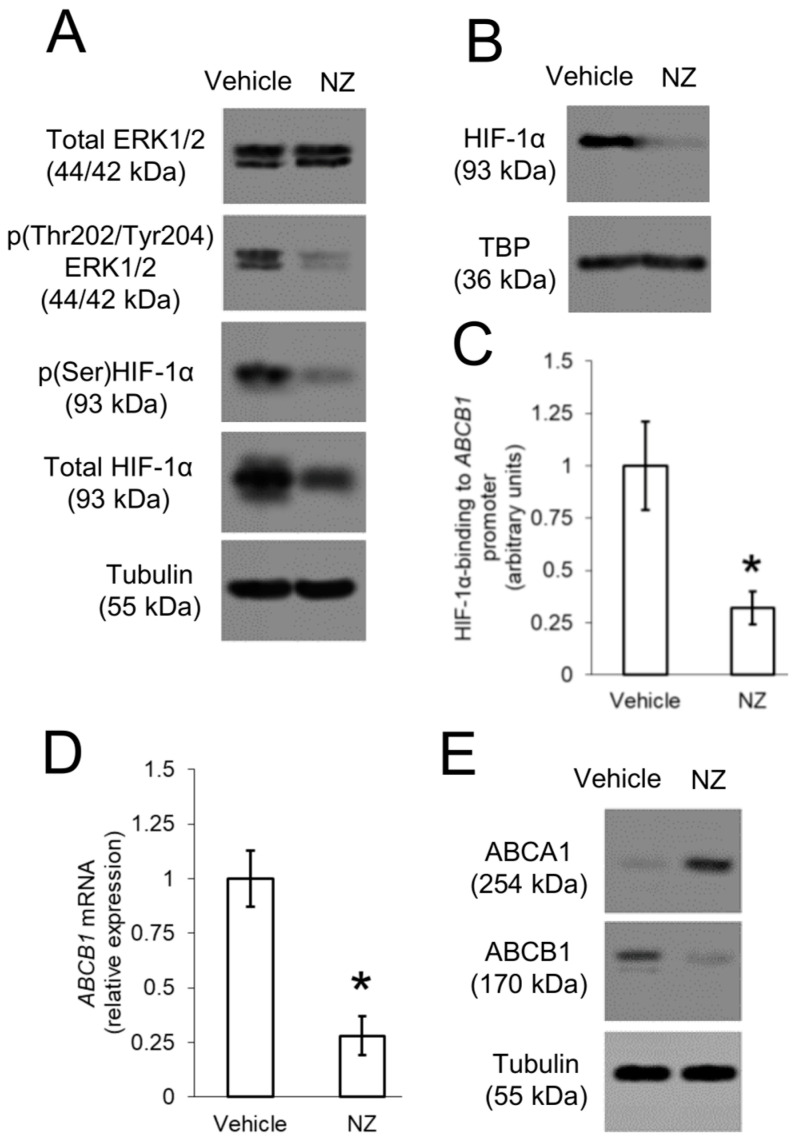
ABCB1 is up-regulated by ERK1/2/HIF-1α axis in resistant osteosarcoma cells. 3D U-2OS cells were treated with sterile physiological solution (vehicle) or self-assembled nanoparticles encapsulating zoledronic acid (NZ), containing 1 μM zoledronic acid for 24 h. (**A**) Immunoblotting of the indicated proteins in whole-cell extracts. The β-tubulin expression was used as control of equal protein loading. The figure is representative of one out of three experiments. (**B**) Immunoblotting of HIF-1α in nuclear extracts. The TBP expression was used as control of equal protein loading. The figure is representative of one out of three experiments. (**C**) HIF-1α binding to *ABCB1* promoter, measured by ChIP in triplicates. Data are means ± SD (n = 3 independent experiments). * *p* < 0.001 for NZ-treated cells vs. vehicle-treated cells. (**D**) ABCB1 mRNA levels, measured by qRT-PCR in triplicates. Data are means ± SD (n = 3 independent experiments). * *p* < 0.001 for NZ-treated cells vs. vehicle-treated cells. (**E**) Immunoblotting of the indicated proteins in whole-cell extracts. The β-tubulin expression was used as control of equal protein loading. The figure is representative of one out of three experiments.

**Figure 5 cells-09-00647-f005:**
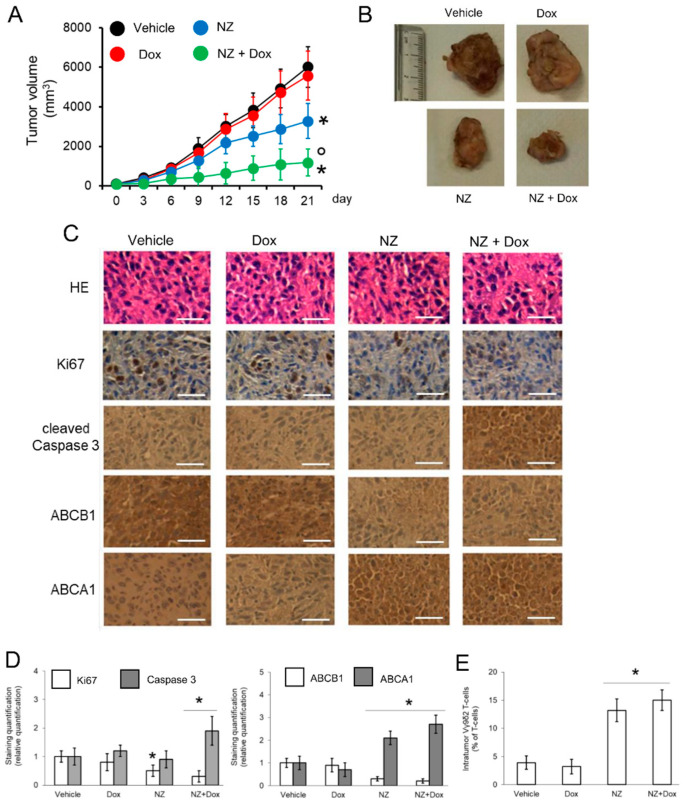
Self-assembled nanoparticles encapsulating zoledronic acid restores chemo-immune-sensitivity in vivo. U-2OS 3D cells (1 × 10^6^) were injected subcutaneously in Hu-CD34^+^ mice. When the tumor reached the volume of 50 mm^3^, animals (n = 8/group) were randomized and treated (on day 3, 9, and 15 after randomization) as it follows: 1) Vehicle group, treated with 0.1 ml saline solution intravenously (i.v.); 2) doxorubicin (Dox) group, treated with 5 mg/kg doxorubicin i.v.; 3) NZ group, treated with 20 μg/mice zoledronic acid i.v.; 4) NZ + Dox group, treated with 20 μg/mice zoledronic acid as NZ and 5 mg/kg doxorubicin i.v. (**A**) Tumor growth was monitored by caliper. * *p* < 0.01 for NZ-treated mice vs. vehicle-treated mice (days 15-21), * *p* < 0.001 for NZ + Dox-treated mice vs. vehicle-treated mice (days 9–21); ° *p* < 0.001 for NZ+Dox-treated mice vs. Dox-treated mice (days 9–21). (**B**) Representative photographs of tumors from each group of treatment. (**C**) Immunohistochemical analysis of tumor slices stained with hematoxylin-eosin (HE) or immunostained for the indicated proteins (63× objective). Bar = 10 μm. The microphotographs are representative of five tumors/each group. (**D**) The amount of Ki67, cleaved caspase 3, ABCB1, and ABCA1 positive cells were calculated using the Photoshop program. The staining intensity of the “Vehicle” group was considered 1. The staining intensity of the other groups was expressed as relative intensity staining vs. Vehicle group. * *p* < 0.01 for NZ-treated/NZ+Dox-treated cells vs. vehicle-treated cells. (**E**) Percentage of intratumor Vγ9δ2 T-lymphocytes vs. all CD3^+^T-lymphocyte-infiltrating cells, measured by flow cytometry. Data are means ± SD (n = 8 animals). * *p* < 0.001 for NZ-treated/NZ+Dox-treated cells vs. vehicle-treated cells.

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
