# Peer review of "ABCA1/ABCB1 Ratio Determines Chemo- and Immune-Sensitivity in Human Osteosarcoma"

_cells, 2020, doi:10.3390/cells9030647_

Round 1

Reviewer 1 Report

The manuscript by Belisario et al. describes an interesting mechanism of combined chemotherapy/immunotherapy-resistance in preclinical models of osteosarcoma, and offers a new therapeutic strategy to overcome this kind of resistance.

The paper is generally well-written and novel. However, there are a few points that should be addressed before publication to enhance the paper and to further support its conclusions:

The authors should explain better the physiological role(s) of gamma/delta T-cells in the introduction section, as well as their role(s) in cancer. Also, the authors should explain why they focused in this study exclusively on this type of T-cells and not alpha/beta T-cells.

Which other types of intratumor immune cells were found in the xenograft tumors (Fig. 4)? Was there a difference in CD4-positive T-cells or B-cells, which may be assessible by immunohistochemical stains for corresponding markers?

All Immunoblot images should have indications of markers for kDa in the figures.

The xenografts depicted in Fig 4a should be further assessed for phenotypic differences between groups. The authors should assess the rates of cell proliferation (e.g. by counting mitoses on H&E stained slides, or by Ki67 stains) and assess the rate of apoptosis (e.g. by stains for cleaved caspase 3) and necrosis (H&E).

The IHC images show in Fig 4b are very hard to interpret due to the strong yellowish background. These images currently do not help to show to the reader the differences in ABCB1/A1 positive cells per group. Thus, the quality of these representative images needs to be enhanced.

Fig 1. could be supported by showing the relative mRNA levels of ABCB1 and ABCA1 in primary osteosarcoma tumors, e.g. by assessment of published gene expression data (e.g. the NCI TARGET osteosarcoma data https://ocg.cancer.gov/programs/target/data-matrix). If both genes are represented on the used mRNA omics-platform, the authors could also check whether the expression of both genes affects patient outcome by crossing the expression data with the matched clinical data (ftp://caftpd.nci.nih.gov/pub/OCG-DCC/TARGET/OS/clinical/harmonized/).

Although the manuscript is generally well-written, it could be enhanced in terms of grammar by proof-reading by a native English speaker.

Author Response

The manuscript by Belisario et al. describes an interesting mechanism of combined chemotherapy/immunotherapy-resistance in preclinical models of osteosarcoma, and offers a new therapeutic strategy to overcome this kind of resistance.

The paper is generally well-written and novel. However, there are a few points that should be addressed before publication to enhance the paper and to further support its conclusions:

1) The authors should explain better the physiological role(s) of gamma/delta T-cells in the introduction section, as well as their role(s) in cancer. Also, the authors should explain why they focused in this study exclusively on this type of T-cells and not alpha/beta T-cells.

As requested we detailed more the physiological and pathological roles of γδ T-cells, their role in cancer progression and the reasons of considering this specific subset instead of αβ T-cells.

We specified that γδ T-cells are less than 10% of circulating T-lymphocytes, but they are a population abundant at tissue levels [Raverdeau M, Front. Immunol. 2018, 9, e2389]. For this reason, they are physiologically involved in the mucosae defense from microbial pathogens, whereas they induce a strong pro-inflammatory response [Shiromizu CM and Jancic CC, Front. Immunol. 2018, 9, e2389]. On the other hand, a prolonged activation of tissue-infiltrating γδ T-cells favors the break of immune tolerance and the onset of autoimmune diseases [Shiromizu CM and Jancic CC, Front. Immunol. 2018, 9, e2389]. γδ T-lymphocytes peculiarly recognize non-major histocompatible complex (MHC) antigens, structurally characterized by a phosphate moieties (phosphoantigens) [Sebeystien Z, Nat. Rev. Drug Discov. 2019], mounting a fast response, with intermediate features between innate and adaptive immune-response [Raverdeau M, Front. Immunol. 2018, 9, e2389; Hayday AC and Vantourout P, Annu. Rev. Immunol. 2020]. Different subset of γδ T-cells, producing different cytokines, have pro-tumor or anti-tumor effect [Raverdeau M, Front. Immunol. 2018, 9, e2389; Sebeystien Z, Nat. Rev. Drug Discov. 2019]. However, it is generally recognized that a high infiltration of Vγ9Vδ2 T-cell subset is a good prognostic factor in solid tumors [Gentles AJ, Nat. Med. 2015, 21, 938-945]. In settings where the anti-tumor efficacy of αβ CD8+T-cells is impaired, activated Vγ9Vδ2 T-cells may represent one of the most relevant population in the immune-eradication of cancer cells. Since ABCB1-expressing tumors are unable to undergo to ICD and CD8+T-cell-killing after chemotherapy with doxorubicin [Salaroglio IC, Oncotarget 2015, 6, 1128–1142; Kopecka J, Oncotarget 2016, 7, 20753–20772], we focused our attention on the possibility to rescue at least part of the immunogenic effect of doxorubicin by increasing the activation of Vγ9Vδ2 T-cells against chemoresistant tumors.

We modified Introduction (line 79) and references accordingly.

2) Which other types of intratumor immune cells were found in the xenograft tumors (Fig. 4)? Was there a difference in CD4-positive T-cells or B-cells, which may be assessible by immunohistochemical stains for corresponding markers?

We immuno-phenotyped the tumor-infiltrating cells, collecting cells from tumor homogenates by centrifugation on Ficoll-Hypaque density gradient and flow cytometry analysis. As shown in the new Supplementary Figure S6, the vast majority of lymphocytic infiltrating tumors were CD3+CD4+T-lymphocytes, followed by CD3+CD8+ T-lymphocytes. There were no significant differences between the groups of treatments. B-lymphocytes were undetectable. This finding is in line with previous observations showing that Vγ9Vδ2 T-cells infiltrating the tumors vary independently from other T-lymphocyte subsets [Tosolini M et al, Oncoimmunology. 2017, 6, e1284723]. Indeed the former recognize different antigens and mount a different immune-response [Sebeystien Z, Nat. Rev. Drug Discov. 2019; Raverdeau M, Front. Immunol. 2018, 9, e2389; Hayday AC and Vantourout P, Annu. Rev. Immunol. 2020].

We modified Materials and methods (line 269), Results (line 450), Discussion (line 566) and Supplementary materials accordingly. We add one new reference.

3) All Immunoblot images should have indications of markers for kDa in the figures.

We added the indications of the markers as kDa in all the immunoblots.

4) The xenografts depicted in Fig 4a should be further assessed for phenotypic differences between groups. The authors should assess the rates of cell proliferation (e.g. by counting mitoses on H&E stained slides, or by Ki67 stains) and assess the rate of apoptosis (e.g. by stains for cleaved caspase 3) and necrosis (H&E).

Following the Reviewer’s suggestion we performed hematoxylin-eosin staining, immunostainings for Ki67, an index of proliferation, and cleaved caspase 3, an index of apoptosis. As shown in the new Figure 5C, the hematoxylin-eosin staining indicated that all tumors have small areas of necrosis that were significantly higher in the NZ + doxorubicin group. Animal treated with vehicle, doxorubicin or NZ showed high positivity for Ki67 and low positivity for cleaved caspase 3. By contrast, the tumors derived from NZ + doxorubicin group showed decreased proliferation and increased apoptosis.

These results indicated that the association of NZ and doxorubicin recapitulates in vivo what it is observed in ex vivo co-culture models, i.e. an increased percentage of necro-apoptotic cells, indicative of Vγ9Vδ2 T-cell killing activity. This feature, together with a decreased proliferation, may explain the marked reduction in tumor growth.

We modified Materials and methods (line 260), Results (line 443), Discussion (line 557) and legend of Figure 5 accordingly.

5) The IHC images show in Fig 4b are very hard to interpret due to the strong yellowish background. These images currently do not help to show to the reader the differences in ABCB1/A1 positive cells per group. Thus, the quality of these representative images needs to be enhanced.

We replaced the immunohistochemistry analysis with new staining, shown in the new Figure 5C. We re-performed the quantification (Figure 5D).

6) Fig 1. could be supported by showing the relative mRNA levels of ABCB1 and ABCA1 in primary osteosarcoma tumors, e.g. by assessment of published gene expression data (e.g. the NCI TARGET osteosarcoma data https://ocg.cancer.gov/programs/target/data-matrix). If both genes are represented on the used mRNA omics-platform, the authors could also check whether the expression of both genes affects patient outcome by crossing the expression data with the matched clinical data (ftp://caftpd.nci.nih.gov/pub/OCG-DCC/TARGET/OS/clinical/harmonized/).

We thank the Reviewer for this suggestion.

In order to accomplish this remark, we decided to use our own data instead of data provided by other centers. In the revised version of the manuscript, we have added our results on ABCA1 and ABCB1 mRNA expression levels obtained on a series of 21 high-grade osteosarcoma samples, in comparison with human normal muscle tissues (Supplementary Table S1) and in relation to clinical outcome (Figure S2). The analysis suggested that a ABCB1highABCA1low phenotype may indicate a worsen prognosis, although there was not a statistically significant difference.

We have therefore inserted a new paragraph in the Materials and Methods (line 230),in the Results (line 309) and in the Discussion (line 492) sections, together with new Supplementary Materials (Table S1; Figure S2). We added three new references.

7) Although the manuscript is generally well-written, it could be enhanced in terms of grammar by proof-reading by a native English speaker.

The grammar was revised by a native English speaker.

Reviewer 2 Report

Manuscript Number: Cells-716009: Reviewer comments

The manuscript entitles « ABCA1/ABCB1 ratio determines chemo- and immune-sensitivity in human osteosarcoma » by Dimas Carolina Belisario, Muhlis Akman et al., presents original data on the importance of the ration ABCA1/ABCB1 in the sensitivity/resistance to doxorubicin and the anti-tumor immune response. This manuscript also establishes that amino-bisphosphonate treatment can reverse the resistance to doxorubicin.

The manuscript is well written and the experimental strategy is adapted to answer the raised questions. The data reports in the manuscript are interesting and the reviewer has only minor comments.

Minor comments:

Lines 32 and 73: “ABCA1-dependent” instead of “ABCA1-depedent”.

May authors give more data on the mechanism of resistance to Doxorubicin associated to ABCB1 high expression and ABCA1 low expression? Is-it linked to the penetration of doxorubicin into the target tumor cell? Is it link to active transport of Doxorubicin or efflux of doxorubicin? This may help the reader to link resistance to doxorubicin to NZ and immune infiltrate.

The reviewer would appreciate the minor comments to be considered.

Author Response

The manuscript entitles « ABCA1/ABCB1 ratio determines chemo- and immune-sensitivity in human osteosarcoma » by Dimas Carolina Belisario, Muhlis Akman et al., presents original data on the importance of the ration ABCA1/ABCB1 in the sensitivity/resistance to doxorubicin and the anti-tumor immune response. This manuscript also establishes that amino-bisphosphonate treatment can reverse the resistance to doxorubicin.

The manuscript is well written and the experimental strategy is adapted to answer the raised questions. The data reports in the manuscript are interesting and the reviewer has only minor comments.

Minor comments:

1) Lines 32 and 73: “ABCA1-dependent” instead of “ABCA1-depedent”.

We corrected the typos.

2) May authors give more data on the mechanism of resistance to Doxorubicin associated to ABCB1 high expression and ABCA1 low expression? Is-it linked to the penetration of doxorubicin into the target tumor cell? Is it link to active transport of Doxorubicin or efflux of doxorubicin? This may help the reader to link resistance to doxorubicin to NZ and immune infiltrate.

We clarified that the active efflux of doxorubicin by ABCB1 is the main mechanism of resistance to this drug in osteosarcoma. As a consequence, doxorubicin is poorly accumulated within osteosarcoma cells and its pleiotropic mechanisms of cytotoxicity are impaired. In chemosensitive cells doxorubicin can elicit a direct killing of cancer cells, by inhibiting topoisomerase II, by increasing reactive oxygen and nitrogen species, by perturbing mitochondria functions [Granados-Principal S, Food Chem. Toxicol. 2010, 48, 1425-1438], and an indirect killing, by increasing the immunogenic cell death (ICD), i.e. the ability of host immune system to eradicate the target cells [Bezu L, Front. Immunol. 2015, 6, e187]. However, both direct killing and ICD are impaired in ABCB1-expressing cells. For this reason, chemo-resistant cells are also immune-resistant. Hence, strategies alternative to the canonical ICD are needed to overcome such immune-resistance. With this goal in mind, we focused on the possible exploitation of Vγ9Vδ2 T-cells, a T-cell subset that plays a key role in anti-tumor immunity [Silva-Santos B, Nat. Rev. Immunol. 2015, 15, 683–691], as an endogenous immune-weapon against ABCB1-expressing osteosarcomas.

We explained that: 1) the endogenous activator of Vγ9Vδ2 T-cells is isopentenyl pyrophosphate (IPP), an isoprenoid metabolite produced during the cholesterol synthesis [Castella B, J. Immunol. 2011, 187, 1578–1590]; 2) ATP Binding Cassette transporter A1 (ABCA1) is the efflux transporter of IPP in antigen presenting cells, bone marrow stromal cells and hematopoietic cells [Castella B, Nat. Commun. 2017, 8, 15663. doi: 10.1038/ncomms15663]; 3) aminobisphosphonates such as zoledronic acid, a strong inhibitor of farnesyl pyrophosphate synthase (FPPS) [Roelofs AJ, Clin. Cancer Res. 2006, 12, 6222s–6230s], increase intracellular accumulation and extracellular release of IPP [Castella B, J. Immunol. 2011, 187, 1578–1590; Castella B, Nat. Commun. 2017, 8, 15663. doi: 10.1038/ncomms15663], activating Vγ9Vδ2 T-lymphocytes endorsed with anti-tumor activity.

Therefore we aim at clarifying if aminobisphosphonates can enhance the release of IPP via ABCA1 and if boosting the ABCA1-dependent activation of Vγ9Vδ2 T-cells could be an effective strategy against ABCB1-expressing osteosarcoma, resistant to doxorubicin.

We hope to have clarified the driving hypothesis of our work, as well as the linkage between the resistance to doxorubicin mediated by ABCB1, the need of using pharmacological strategies based on zoledronic acid to increase ABCA1 and IPP-mediated activation of Vγ9Vδ2 T-cells.

We modified the Abstract (lines 30 and 32), the Introduction (line 56) and the references accordingly.

The reviewer would appreciate the minor comments to be considered.

Round 2

Reviewer 1 Report

The authors address all my concerns.